# Arbuscular Mycorrhizal Fungi and Rhizobium Improve Nutrient Uptake and Microbial Diversity Relative to Dryland Site-Specific Soil Conditions

**DOI:** 10.3390/microorganisms12040667

**Published:** 2024-03-27

**Authors:** Rosalie B. Calderon, Sadikshya R. Dangi

**Affiliations:** Agricultural Research Service, Northern Plains Agricultural Research Laboratory, USDA, 1500 N Central Avenue, Sidney, MT 59270, USA

**Keywords:** field pea, arbuscular mycorrhizal fungi (AMF), Rhizobium, microbial inoculants, root symbiosis, microbiome, agroecosystem, drylands

## Abstract

Arbuscular mycorrhizal fungi (AMF) and rhizobium play a significant role in plant symbiosis. However, their influence on the rhizosphere soil microbiome associated with nutrient acquisition and soil health is not well defined in the drylands of Montana (MT), USA. This study investigated the effect of microbial inoculants as seed treatment on pea yield, nutrient uptake, potential microbial functions, and rhizosphere soil microbial communities using high-throughput sequencing of 16S and ITS rRNA genes. The experiment was conducted under two contrasting dryland conditions with four treatments: control, single inoculation with AMF or Rhizobium, and dual inoculations of AMF and Rhizobium (AMF+Rhizobium). Our findings revealed that microbial inoculation efficacy was site-specific. AMF+Rhizobium synergistically increased grain yield at Sidney dryland field site (DFS) 2, while at Froid site, DFS 1, AMF improved plant resilience to acidic soil but contributed a marginal yield under non-nutrient limiting conditions. Across dryland sites, the plants’ microbial dependency on AMF+Rhizobium (12%) was higher than single inoculations of AMF (8%) or Rhizobium (4%) alone. Variations in microbial community structure and composition indicate a site-specific response to AMF and AMF+Rhizobium inoculants. Overall, site-specific factors significantly influenced plant nutrient uptake, microbial community dynamics, and functional potential. It underscores the need for tailored management strategies that consider site-specific characteristics to optimize benefits from microbial inoculation.

## 1. Introduction

Dryland farming accounts for 60% of the total crop production, which plays a critical role in feeding the world’s growing population [1,2]. Over 40% of the global dryland area is utilized as cropland, often characterized by limited water availability [1,2]. Other major challenges are poor soil quality due to low organic matter content, nutrients, and biological activity [3]. In resource-limited semi-arid regions like Montana, it is imperative to adopt crop management that promotes soil health, which has distinct effects on microbial populations involved in water use efficiency, nutrient cycling, and other critical soil ecosystem functions.

Pulse crops like field peas are specialty high-value crops integrated into cereal crop rotations because of their ecological and economic benefits. Pea (*Pisum sativum* L.) is one of the most valuable legume crops in the world, with over 14.2 million tons of production value [4]. US pea production is around 873,800 tons, of which 362,119 tons were produced in Montana with a $94 M production value [5]. However, the stable production of peas is hampered by various abiotic and biotic stresses. Peas establish symbiotic associations with beneficial soil microorganisms and provide 3–20% photosynthetically assimilated carbon (C) to AMF and Rhizobium partners [6]. As a leguminous plant, it can obtain over 80% of its nitrogen (N) requirement from biological N fixation [7]. Further, its mycorrhizal associations enhance water and nutrient uptake, including fixed N by bacteria and phosphorus (P) at significantly higher rates than nonmycorrhizal plants [6,7,8,9]. These microbial symbioses impose a high demand for carbon from plants, requiring increased CO_2_ assimilation from the atmosphere [10], thereby increasing plant exudates, residue, and a microbial biomass that enhances soil organic C storage [11].

Agricultural practices heavily relying on fertilizers and pesticides can deplete soil carbon and fertility, decrease pH, and reduce soil microbial diversity and plant vigor [10,11,12]. Using beneficial microbes is considered a tool to circumvent agricultural intensification [13,14]. Microorganisms have been utilized as biofertilizers, biocontrol agents, and biostimulants over decades. Several studies have reported the successful use of microbial inoculants, increasing crop yield and plant fitness [13,14,15,16,17], while others have reported nil to varying efficacy in field conditions [18,19]. In particular, arbuscular mycorrhizal fungi (AMF) have been recognized for their C, N, and P cycling, soil structure improvement, water and nutrient uptake, and endowing plant tolerance to abiotic and biotic stresses [12]. Rhizobium and other beneficial soil microorganisms enhance plant tolerance to biotic stress through Induced Systemic Resistance (ISR) [20] and environmental stresses through the production of the enzyme 1-amino-cyclopropane carboxylate deaminase (ACCD), which decreases the stress ethylene level by breaking down the ethylene precursor ACC into 2- oxobutanoate and ammonium (NH_4_), facilitating plant growth [21]. Harnessing these beneficial traits could offset plant abiotic and biotic stresses, reducing reliance on agrochemical input. However, the influence of AMF and Rhizobium as microbial inoculants on crop agronomic performance is not well resolved in dryland field conditions.

Leguminous and mycorrhizal plant symbioses with Rhizobium and AMF have been known as a tripartite interaction. However, the plant-soil microbiome interaction is considered highly complex, involving the plant host, microbial communities, soil, and associated environmental conditions contributing to soil health and productivity [15,16]. Studies of natural populations have shown that groups of microbes with distinct functions are crucial in nutrient mineralization, degradation of organic residues, and nutrient availability for plant utilization [16,19]. Recent studies have shown that AMF, phosphate-solubilizing bacteria, and the plant microbiome contribute to nutrient transformation and improved yield in other economically important crops such as barley [22] and corn [23]. Further, the rhizosphere soil microbial communities are influenced by plant exudates, soil-physicochemical properties, and management practices [24,25,26]. The microbial inoculants, when applied as seed treatments, may have a ‘priority effect’ in the microbiome assembly and function in the early plant developmental stage [27,28]. Understanding the rhizosphere soil microbial communities’ response to microbial inoculants, dryland crop management practices, and site-specific conditions can provide insights into the essential functions contributing to plant fitness and resilience. Currently, there is a limited understanding of the influence of microbial inoculants as pea seed treatment on bacterial and fungal communities relative to soil health in dryland agroecosystems.

To better understand the influence of AMF and Rhizobium inoculation on plant growth and soil health at contrasting dryland sites, we used high-throughput sequencing of the 16S and ITS rRNA gene amplicons to elucidate the bacterial and fungal communities and their potential ecophysiological functions. The specific objectives of this research were to determine the influence of microbial inoculants on pea growth and nutrient dynamics; microbial diversity, structure, and composition; and potential microbial metabolic and ecophysiological functions. We hypothesize that microbial inoculation and dryland site-specific conditions significantly affect the pea rhizosphere microbial community, affecting crop productivity and soil health.

## 2. Materials and Methods

### 2.1. Site and Soil Characterization

This study was conducted in contrasting dryland field sites, in Froid (48 C15′18.972″ N, 104°29′39.843″ W) and Sidney (48°15′18.972″ N, 104°29′39.843″ W), eastern Montana (Appendix A). The soil is characterized as Dooley sandy loam in Froid dryland field site (DFS) 1, while DFS 2 in Sidney is Williams loam soil, as previously described by Sainju et al. 2022 [29]. DFS 1 had strongly acidic soil (pH = 4.7) and low soil organic matter, while DFS2 had a slightly neutral pH (pH = 6.33) and moderate soil organic matter. Both sites have a mean air temperature of 21 °C. The precipitation during the growing season (May to August 2022) was 55 mm and 25 mm for DFS 1 and DFS 2, respectively (Appendix A). DFS 1 has an available water supply of 0.17 cm of water per cm of soil, while DFS 2 has an available water supply of 0.18 cm per cm of soil (Appendix A, USDA Soil Survey). Soil samples were collected at each site and analyzed for basal soil physicochemical properties. Soil analysis was performed at the Ward laboratories, Kearney, NE, following the prescribed protocol [30,31,32]. Briefly, soil N was extracted using 5 g soil in 15 mL of 1 M potassium chloride solution; soil P using 2 g soil in 20 mL Mehlich solution; and soil K using 1 g soil in 10 mL of 1 N ammonium acetate (NH_4_OAc). The soil nutrients in the soil-filtered extract were measured by flow injection analysis. Both field sites were previously planted each year with wheat for five years.

### 2.2. Microbial Inoculants

The microbial inoculants specific for field peas were arbuscular mycorrhizal fungi (AMF) *Glomus intraradices* (syn. *Rhizophagus irregularis*), *Rhizobium leguminosarum* bv. *viciae* and a dual inoculant of endomycorrhizal (AMF) and rhizobial products (AMF+Rhizobium) commercially produced by Premier Tech (Riviere-du-Loup, QC, Canada). The active ingredients were 2750 viable spores of *G. intraradices*/g and 1.6 × 10^9^ viable cells of *R. leguminosarum* bv. *viciae* per gram of the product. These inoculants in peat-based powdered form were applied as seed treatment following the recommended 300 g/ha application rate for peas at a 224 kg/ha seeding rate [33].

### 2.3. Dryland Field Experiment

Pea seeds (*Pisum sativum* L., forage variety 4010) inoculated with microbial inoculants were planted in the dryland field sites described above. The study was set up in 1858 m^2^, subdivided into five blocks of equal size, and each block had four plots. Each plot had a 6.10 m width × 15.24 m length. There were four treatments: control, AMF, Rhizobium, and dual inoculants (AMF+Rhizobium) arranged in a randomized complete block design (RCBD) with five replications in each site. All field sites were managed under no-till and regular cultural practices for field peas (Appendix A). Monoammonium phosphate and muriate of potash basal fertilizer were applied at seeding at 56 kg/ha and 45 kg/ha, respectively. Basagran 5 L herbicide was applied at an 897 g/ha rate to control weeds at the vegetative stage. Field pea plots were planted on 15 May (DFS 1) and 17 May 2022 (DFS 2). Before harvesting, biomass and plant yield components were gathered. The pea grains were machine harvested at maturity on 19 August (DFS 1) and 9 August 2022 (DFS 2) using a combine harvester.

### 2.4. Plant Nutrient Uptake, Yield, and Microbial Dependency

The plant biomass and grain were ground separately to determine the nutrient concentration. A 0.29 to 0.30 g weight per sample was placed into a ceramic boat. C and N concentrations were measured using the LECO FP-2000 C-N analyzer (LECO Trumac Series, LECO Corporation, St. Joseph, MI, USA). P content in the pea grain was measured with an inductively coupled plasma (ICP) after hot block digestion following previously described methods [34]. Nutrient uptake and carbon sequestration were calculated with the formula: nutrient uptake or carbon sequestration = (grain yield kg/ha × nutrient concentration) + (biomass yield kg/ha × nutrient concentration) [35]. Total residual nutrient (nitrate, phosphorus) in the 0–60 cm depth soil was calculated with the formula: total residual = nutrient concentration in ppm (mg/kg) × mass of soil (kg/ha), where mass of soil (k/ha) = 100 (m) × 100 (m) × soil depth (m) × soil bulk density × 1000 (1000 kg = 1 t conversion factor). Microbial dependency (MD) of inoculation treatment was calculated according to Van Der Heijden 2002 [36] as follows: MD% = [(T − C)/C] × 100, where T is the mean plant growth parameters (biomass or yield) in the given replicates of the microbial inoculated treatment, and C is the mean in the corresponding noninoculated treatment or control group. Microbial dependency with positive values indicates plant growth promotion, and negative values indicate plant growth suppression by microbial inoculants.

### 2.5. Microbial Community Analysis

We explored bacterial and fungal populations by MiSeq sequencing, targeting the 16S and ITS rRNA gene amplicons, respectively. Forty-two days after seeding at the pea vegetative stage, rhizosphere soils closely attached to the roots of three plants from each treatment were composited as a sample for each treatment per plot. Rhizosphere soil sample collections were conducted on 29 June for DFS 1 and 30 June 2022 for DFS 2. Genomic DNA of rhizosphere soil fraction was extracted using a PowerSoil kit (Qiagen, Valencia, CA, USA) following the manufacturer’s instructions. The extracted DNA samples were analyzed for quality and quantity using NanoDrop 1000 (Thermo Scientific, Wilmington, DE, USA). The DNA samples were sent to the University of Minnesota Genomics Center Microbiome Services for sequencing following the 16S and ITS Illumina Amplicon Protocol. The V4–V5 region of the 16S rRNA gene was amplified with the 515F/926R primer pair [37,38], while the fungal ITS region was amplified using ITS1F/ITS2_Nextera primer pairs [39,40]. The PCR product was diluted 1:100 with molecular-grade water. The diluted amplicons were indexed using barcoded PCR primers [40]. Barcoded amplicons were normalized using SequalPrep kits (Invitrogen, Carlsbad, CA, USA). The normalized libraries were cleaned with AMPure XP mag beads (Beckman Coulter, Brea, CA, USA), quantified by Qubit, and sequenced using the MiSeq v3 600-cycle kit on the MiSeq platform (Illumina, San Diego, CA, USA).

Quantitative Insights into Microbial Ecology (QIIME 2 2023.3) bioinformatic pipeline was used to analyze the raw sequences [41]. 16S rRNA demultiplexed paired-end fastq files were demultiplexed with adapters removed in the process, while the primer and adapter sequences of ITS demultiplexed paired-end fastq files were trimmed off at the 5′ and 3′ end, respectively, using cutadapt [42]. DADA2 was used for denoising, sequence correction, and removal of chimeras (qiime dada2 denoise-paired with the following parameters: –p-trim-left-f 0 paired –p-trim-left-r 0 –p-trunc-len-f 290 –p-trunc-len-r 290 for 16S rRNA sequences, while –p-trim-left-f 0 paired –p-trim-left-r 0 –p-trunc-len-f 170 –p-trunc-len-r 170 was used for the ITS sequences) [43]. The forward and reverse reads of all the sequences were truncated based on the quality scores (phred ≥ 30).

The BIOM table was summarized with the Qiime2 ‘feature table summarize’ command. A phylogenetic tree was constructed using qiime phylogeny align-to-tree-mafft-fasttree [44,45]. The amplicon sequence variant (ASV) taxonomic identification was performed by using the q2-feature-classifier [46] against the SILVA 132 [47] database and BLAST+ consensus taxonomy classifier [48] for prokaryote 16S rRNA genes, while the UNITE v8 database [49] and classify sklearn were used for fungal ITS. We obtained 1,474,948 total 16S rRNA sequences, of which 162,466 were prokaryotes with high-quality reads clustered into 2862 amplicon sequence variants (ASVs). Of the 1,264,985 ITS sequences, 750,118 were high-quality fungal reads clustered into 2190 ASVs. Data were filtered for low counts of 20% prevalence with a minimum of four counts in each sample and a 10% low variance filter based on the interquartile range. All samples were rarified to even sequencing depth based on the lowest sampling depth. Principal coordinate analysis (PcoA) was used to visualize the effect of microbial inoculants on the microbial community composition and structure [50]. Microbiome datasets with metadata were further visualized using microbiomeAnalyst [51] and RAWgraphs [52].

The Tax4Fun2 R package was used to estimate the metabolic functional features of bacterial communities, which integrates data from 16S rRNA genes with the Kyoto Encyclopedia of Genes and Genomes [53]. We focused on predicted genes involved in C fixation, N metabolism, and P cycling. Fungal ecological functions were predicted using the database FungalTraits [54], focusing on the ecological activities of fungi relative to nutrient cycling, plant-microbe dynamics, and soil health.

The influence of microbial inoculants on rhizosphere microbial communities in the dryland sites was examined using permutational multivariate analysis of variance (PERMANOVA) [55]. The heat tree analysis was used to plot the taxa differential abundance relative to high grain yield in the AMF microbial treated in DFS1 and the AMF+Rhizobium treated in DFS2 in comparison with the untreated control based on median abundance and non-parametric Wilcoxon test [56]. Two-way ANOVA was conducted using JMP Pro Statistics, version 17 (SAS Institute, Cary, NC, USA) to test the effects of microbial inoculants and dryland site conditions on nutrient content, nutrient uptake, and yield. The normal distribution and the variance homoscedasticity were analyzed using Shapiro–Wilk and Levene’s tests. Non-parametric Kruskal–Wallis test analysis was performed on the variables that failed the test. Post hoc mean comparisons were completed with Protected Fisher’s Least Significant Difference (LSD).

## 3. Results

### 3.1. Influence of Microbial Inoculants on Plant Agronomic Performance and Nutrient Dynamics

Pea grain yield was significantly affected by AMF+Rhizobium in the dryland field site 2 (DFS 2) with moderate soil organic matter (SOM) and neutral pH (Figure 1a). Plants inoculated with AMF+Rhizobium had a significantly higher yield than the control (z = 2.506, *p* = 0.012). Based on microbial dependency analysis, a positive contribution of AMF+Rhizobium (16%) and a negative contribution of AMF treatment (−8%) were linked to yield and overall plant growth under DFS 2 (Figure 1b), while in DFS 1, with low SOM and soil pH, AMF treatment contributed 25% to plant growth and yield; nonetheless, there was no significant yield difference among treatments at this site (Figure 1b). DFS 2 had a significantly higher harvested grain yield than DFS 1. The high initial soil nitrogen did not translate to a higher grain yield, although a more profuse vegetative plant growth was observed in DFS 1 (Appendix A). Across dryland sites with high soil P fertility levels, the microbial inoculants’ contribution to the overall plant growth and yield was higher with AMF+Rhizobium (12%) than with single inoculations of AMF (8%) or Rhizobium (4%) (Figure 1b and Appendix A). No significant variations among the treatments were observed in plant stand, nodulation, and plant biomass (Appendix A).

The initial available soil nutrients were compared to the pea nutrient requirements (Appendix A). DFS 1 had a strongly acidic soil (pH = 4.7), low organic matter content (1.5% LOI), medium nitrogen (28.33 ppm N), very high phosphorus (73.00 ppm P), and potassium (214.5 ppm K). Meanwhile, DFS 2 had a slightly neutral pH (pH = 6.33), moderate organic matter content (2.5% LOI), low nitrogen (7.60 ppm N), high phosphorus (44.75 ppm P), and very high potassium (277.87 ppm K). The nutrient contents in the grains were less influenced by microbial inoculants but significantly differed between dryland sites. Plants treated with AMF+Rhizobium and AMF inoculants had a higher percentage of carbon (% C) in the grains than the control in DFS 1 (*p* = 0.04, Figure 2a,b and Appendix A). Carbon content (kg/ha) in plant biomass and grains was significantly higher in DFS 2 than in DFS 1 (Appendix A). There was no significant difference among treatments on N and P uptake within each site (Figure 2c–f). Nonetheless, the trend showed an increased nitrogen and phosphorus content in pea grains of AMF+Rhizobium and AMF treatments, respectively. The nitrogen content in the grains was significantly higher in DFS 1 with greater initial soil N than in DFS 2. Phosphorus content in the grains was significantly higher in DFS 2, although it had a lesser initial soil P than in DFS 1 with strongly acidic soil (Appendix A). The effect of site-specific conditions was greater than that of microbial inoculants on plant nutrient uptake (Appendix A). DFS 2 had a significantly higher total plant carbon and P uptake between sites, while DFS 1 had a significantly higher N uptake irrespective of the microbial inoculants (*p* ≤ 0.001).

The initial soil nitrogen was significantly high in DFS 1, resulting in higher nutrientresiduals than in DFS 2 (Figure 3a and Appendix A). After the pea cropping, the soil N residual was significantly higher than the nutrient utilized by the plants in DFS 1. In DFS 2, plants utilized more nutrients efficiently, thus, there were less N residuals (Figure 3c). Soil P residuals were significantly higher in DFS 1 than in DFS 2. Across sites, soil P residual was significantly higher than grain P uptake (Figure 3d). The difference in nutrient dynamics indicates a more significant influence of site-specific variation than microbial inoculation, which influences nutrient availability and plant uptake beyond initial soil nutrients.

### 3.2. Variation in the Soil Microbial Communities Associated with Increased Plant Yield in Two Contrasting Dryland Sites

Microbial alpha-diversity analysis in dryland field sites displayed various levels of richness, as indicated by the number of species observed (Figure 4a,b). AMF+Rhizobium and the control exhibited the highest bacterial diversity, while AMF-treated plants in the DFS 1 showed the lowest mean number of bacterial species (Figure 4a). No significant difference was observed in bacterial species richness between sites (Appendix A and Appendix A). On the other hand, fungal species richness significantly increased in the rhizosphere of plants treated with microbial inoculants (Figure 4b). DFS 2 had significantly higher fungal species richness than DFS 1 (*p* < 0.001). The Bray–Curtis index was used to calculate beta-diversity values visualized in PCoA plots, and the PERMANOVA significance test was used to determine the variations in microbial community structures. Clustering of samples based on the site and site-specific response to microbial inoculants significantly explained (*p* < 0.001) the highest percentage of variation in community structure for the bacterial fraction (R^2^ = 0.31 and R^2^ = 0.43) (Figure 4c,d) and fungal (R^2^ = 0.48 and R^2^ = 0.55) fractions of the population (Figure 4e,f). These data showed that specific sites favor distinct soil microbial communities and a more prevalent site-specific response to microbial inoculants. Site-specific conditions strongly influenced the microbial community structure, while a minimal variance was attributed to microbial inoculants.

The compositional differences of rhizosphere soil microbiomes as influenced by site-specific response to microbial inoculants were shown in the distribution of the dominant phyla and taxonomic orders (Figure 5, Appendix A). Of note is the significant (*p* < 0.01) enrichment of Actinobacteria (relative abundance ~35–38%) in the bacterial communities in both dryland sites (Figure 5a). Actinobacteria was significantly high in Rhizobium-treated microbial communities. By comparison, DFS 2 soils contained the highest percentage of Bacteroidetes, Firmicutes, and Nitrospirae (Appendix A), while WPS2 and Planctomycetes were enriched in DFS 1. These dominant phyla were represented by 11 abundant orders (>1% of all sequences). Of these, seven taxonomic orders, which include *Propionibacteriales* (relative abundance ~5–19%), *Rhizobiales* (relative abundance ~5–7%), *Chitinophagales* (relative abundance ~5–7%), *Frankiales* (relative abundance ~3–7%), *Elsterales* (relative abundance ~2–6%), *Thermomicrobiales* (relative abundance ~0.3–6%), and *Bacillales* (relative abundance ~0.8–5%) were significantly enriched in DFS 2 (Figure 5c and Appendix A); meanwhile, Acetobacterales, Acidobacteriales (relative abundance ~0.5–6%), Gaiellales (relative abundance ~6%), and uncultured bacterium (relative abundance ~9%) were enriched in DFS 1 (Figure 5c). *Rhizobiales* was more abundant in the microbial treatments than in the control in DFS 2. Rhizosphere soil of peas treated by AMF+Rhizobium had significant enrichment of *Elsterales*. AMF-treated microbial communities were enriched in Acidobacteriales (Appendix A). *Elsterales, Nitrospirales, Nitrososphaerales, Thermomicrobiales, Frankiales*, *Betaproteobacteriales, Chitinophagales,* and *Bacillales* were abundant in DFS 2 compare to DFS 1 (Figure 5c).

Moreover, fungal community patterns were more delineated by sites. Mortierellomycota, Chytridiomycota, Glomeromycota, and unassigned phyla were enriched in DFS 2, while Mucoromycota was enriched in DFS 1 (Figure 5b and Appendix A). Significant enrichment of taxonomic orders of Mortierellales, Pleosporales, Coniochatales, Helotiales, Chaetosphaeriales, and Glomerales were observed in DFS 2, while Eurotiales, Filobasidiales, Holtermanniales, Rhizophlyctidales, Sordariales and Tremellales were observed in DFS 1 (Figure 5d and Appendix A). The fungal composition data showed that site-specific factors strongly influence the fungal communities in the rhizosphere, wherein each site harbors a unique set of fungal taxa.

Taxonomic hierarchical data comparison based on the heat tree analysis of the microbial communities associated with high grain yield in the AMF-treated plants in DFS 1 revealed a lesser abundance of *Chujaibacter* and *Segetibacter* bacteria than in the untreated control (Figure 6a and Appendix A). In contrast, fungal communities in the AMF-treated plants had a significantly higher *Selenophoma* under the order Dothideales, *Keissleriella,* and *Parastagonospora* under the family Lentitheciaceae within the fungal order Pleosporales than the untreated control (Figure 6b and Appendix A). The decreased bacteria and increased abundance of fungal groups indicate a community shift in response to the AMF treatment, favoring some fungal over bacterial groups in challenging conditions. Meanwhile, microbial communities in the AMF+Rhizobium-treated plants associated with significantly high yield in DFS 2 revealed an enrichment of several uncultured bacteria under Elsterales (Figure 6c and Appendix A). Further, *Exophiala*, *Mortierella, Fusicolla,* and other fungal species under the Sordariomycetes were enriched in the AMF+Rhizobium treatment than the untreated control (Figure 6d and Appendix A). The differentially abundant taxa indicate the influence of AMF and AMF+Rhizobium on the microbial communities, contributing to increased plant yield in unfavorable and non-P limiting conditions, respectively.

### 3.3. Effect of Microbial Inoculants on the Potential Functions of the Microbial Community Relative to Nutrient Cycling and Soil Health

The predicted metabolic functional profile of bacterial communities was obtained using Tax4Fun2 based on the 16S rRNA genes and the KEGG Ortholog groups (KOs). We narrowed down the predicted genes to C, N, and P nutrient cycling (Figure 7a and Appendix A). The relative abundance of C and N genes was significantly higher in DFS 2 than in DFS 1 (*p* ≤ 0.001), while P uptake, P solubilization, and P starvation regulation did not vary between sites with very high P. Investigating fungal ecophysiological functions revealed significant differences between dryland sites, indicating that each site’s fungal community has distinct ecological roles and functional adaptations. DFS 2 had a higher relative abundance of arbuscular mycorrhizal fungi (AMF), biocontrol agents, plant pathogens, and others, while DFS 1 had a higher relative abundance of saprotrophs and mycoparasites (Figure 7b and Appendix A). In non-P nutrient or other limiting conditions, microbial community functions relative to nutrient cycling and ecological functions were more influenced by site-specific factors than microbial inoculations.

## 4. Discussion

In this study, the effectiveness of microbial inoculants exhibited variation across contrasting dryland sites. AMF+Rhizobium contributed positively to plant growth and yield in non-P nutrient-limited but low N conditions. The synergy of AMF and Rhizobium had been reported in previous studies in other legumes such as soybean (*Glycine max*) [57] and alfalfa (*Medicago* spp.) [6]. Conversely, AMF treatment had a negative impact on plant growth and yield in dryland sites with high P fertility levels. These findings corroborate previous studies that AMF does not further improve the host plant’s P budget at high P levels in the rhizosphere [4,58,59]. Across dryland sites, the marginal effect of microbial inoculants on grain yield suggests that the net costs of the symbiosis exceed the benefits for the plants when nutrients are abundant. Plants tend to rely on direct mineral uptake independent from beneficial microbes because direct uptake requires less energy than the demanding process of microbial-dependent uptake [60]. In our study, high initial soil fertility corresponds to high N and P soil nutrient residuals.

Under unfavorable soil conditions with low organic matter and pH, a positively high microbial dependency on AMF contributed to plant growth and yield. Nonetheless, the yield in DFS 1 was significantly less than in DFS 2, which follows the principles of limiting factors that dictate the level of crop production, which cannot exceed what is allowed by the maximum limiting factor [61]. In this study, soil acidity resulted in nodulation failure irrespective of microbial inoculants. The acidic soil conditions affect plants and microbial symbionts and restrict nutrient availability [62]. Rhizobium symbiosis, which demands high phosphorus (P), faces challenges in acidic soils where P fixation or immobilization occurs, leading to reduced P bioavailability [63]. The high initial soil nutrients in DFS 1 did not translate to a higher yield, indicating that soil pH, soil organic matter (SOM), and other factors substantially influenced yield.

Noteworthy, plants treated with AMF and AMF+Rhizobium had higher carbon content in the grains than the untreated control in DFS 1, suggesting that microbial inoculants may directly influence carbon accumulation in the grains by stimulating plants to acquire more C for the symbioses [10]. However, single inoculations (AMF or Rhizobium) showed a slightly lower C grain than the dual inoculated plants, which points to the plant cost of C allocation being higher than the nutrients gained from AMF or Rhizobium symbiosis alone [64]. Microbial inoculation may indirectly affect C and other nutrient acquisition, leading to plant productivity by enhancing the soil microbial diversity and enriching microbial groups involved in nutrient cycling. Our results showed plants had a higher microbial dependency to acquire water, P, and N resources from AMF+Rhizobium than single inoculations. These findings point to the significance of dual microbial inoculants in promoting carbon sequestration and efficient resource acquisition by the plants in dryland conditions.

Crop productivity and soil health are associated with belowground microbial diversity [12,13,64,65]. Microbial diversity would likely be low in DFS 1 with low soil organic matter, which has less energy sources for the microbes, and the low pH would select for fewer microbial species tolerant to acidic soil conditions. However, our 16S rRNA microbiome data revealed no significant variation in bacterial species richness between the dryland sites, which suggests that bacterial species have a broad adaptation to varying soil conditions, and distinct bacterial groups thrive in each site. The fungal communities showed a more pronounced site-specific response to soil conditions and microbial inoculants. Fungal diversity was higher in the site with favorable soil conditions (DFS 2) and responded more strongly to AMF and AMF+Rhizobium inoculations. Of important note, AMF and AMF+Rhizobium treated in DFS 1 were comparable to the fungal species richness in DFS 2, which points to possible stress alleviation by microbial inoculants. The site-specific response to microbial inoculants was more pronounced for the fungal communities than the bacterial communities, which supports earlier reports that fungal communities tend to be more sensitive to different agronomic practices and environmental changes due to their complex life cycles and niche requirements than bacterial communities [66,67]. AMF+Rhizobium enriched fungal species at different dryland gradients.

In our study, the microbial community structure and composition variation indicates that the specific site favors distinct soil microbial communities and a more prevalent site-specific response to AMF and AMF+Rhizobium microbial inoculants. The bacterial phylum Actinobacteria was enriched in both dryland sites, consistent with another study that this phylum is well adapted to water-limited or drought conditions and likely contributes to other microbial changes in the soil [68]. At DFS 2, the bacterial communities were dominated by Bacteroidetes [69] and Nitrospirae [70] phyla, which are involved in nutrient cycling. Firmicutes was associated with high P soils [71], plant growth promotion, and pathogen suppression [72]. Microbial inoculations with AMF, Rhizobium, and AMF+Rhizobium were enriched in Rhizobiales. This taxonomic order is comprised of mostly nitrogen-fixing bacteria [73]. AMF+Rhizobium treatment significantly enriched Esterales and Frankiales, which suggests a synergistic influence on the abundance of these copiotrophic bacteria thriving in nutrient-rich substrates [74]. AMF, particularly *Rhizophagus irregularis* (syn. *Glomus intraradices*), has been reported to harbor bacterial communities in the hyphosphere of the fungus and increase bacterial resilience in water-limited conditions [75]. At DFS 1, the bacterial communities were dominated by Acidobacteria, a phylum comprised of oligotrophic bacteria that breaks down dissolved organic matter [76], persists in low pH [77], and is associated with low nutrient soil conditions [71]. The other enriched phyla in DFS 1 were Planctomycetes and WPS2. The taxonomic order Acidobacteriales, under Acidobacteria, was significantly enriched in the AMF-treated plant rhizosphere. These findings suggest that soil conditions and microbial inoculants substantially influence the composition of bacterial communities in the rhizosphere.

In the fungal community, Ascomycota and Basidiomycota were the dominant fungal phyla across the sites, but only the composition of other major phyla differed significantly between the sites. At DFS 2, the fungal communities were dominated by saprophytic phyla, Chytridiomycota, and Mortierellomycota [76], and unassigned phylum. In contrast, at DFS 1, the fungal communities were dominated by Mucoromycota phylum with a wide range of beneficial and pathogenic members [78]. There was a significant enrichment of saprophytic and mycoparasitic fungal orders in DFS 2 compared to DFS 1. Some patterns of more saprophytic and mycoparasitic fungal orders were more abundant in the microbial inoculated plants, such as Orbillales, Cystobasidiales, and Myrmecridiales in DFS 1, while Mortieralles and Chaetosphaeriales were more abundant in DFS 2 [54], unlike in the untreated control, with more fungal groups having both saprotrophic and pathogenic life strategies, such as Agaricales, Pezizales, Xylariales, Rhizophlyctidales, and Dothideales [54].

Microbial community dynamics linked with improved crop performance in challenging conditions showed a shift in composition in response to the AMF treatment. AMF promoted the growth of saprophytic fungal genera *Selenophoma* and *Keissleriella* [54], which may compensate for the absence of putative C-cycling bacteria Chujaibacter [79] and other nutrient-cycling bacteria such as Elsterales (Alphaproteobacteria), *Segetibacter* (Bacteroidetes) [69], and uncultured Chloroflexi bacterium in DFS 1. On the other hand, the microbial communities in the AMF+Rhizobium-treated plants associated with a significantly higher yield in DFS 2 had a higher abundance of several uncultured bacteria under Esterales. These bacteria have not been characterized, but they may be beneficial in promoting plant growth and yield. The fungal communities in the AMF+Rhizobium-treated plants had a higher abundance of fungal genera *Exophiala*, *Mortierella,* and *Fusicolla.* These fungi are soil saprotrophs, fungal decomposers, and mycoparasites [54], which may positively affect soil microbes and plants. These microbial changes may also be slightly attributed to the carrier of the commercial microbial inoculants formulated in peat-based powder with high organic matter, which favors wide arrays of microorganisms, including other soil beneficial microorganisms [80]. The enrichment of these beneficial bacteria and fungi in the AMF and AMF+Rhizobium-treated plants may represent positive interactions on possible nutrient exchanges or protection from environmental stress that contribute to increased plant yield.

Potential functions of the microbial community based on the relative abundance of the predicted genes involved in nutrient cycling (C fixation, N metabolism, and P cycling) and ecophysiological traits related to overall soil health revealed a stronger influence of site-specific factors than the microbial inoculants under non-P limiting and unfavorable soil conditions. This indicates that site-specific factors, such as soil pH, organic matter, and nutrient availability, significantly influence the soil microbial communities [26,80,81] and the microbial inoculants. Based on potential functions, fungal communities in each site have distinct ecological functions. Fungal communities in DFS 2 were more likely involved in nutrient cycling and plant-microbe interaction due to a higher relative abundance of AMF, biocontrol agents, and plant pathogens. On the other hand, fungal communities in DFS 1 were more likely involved in decomposition and nutrient cycling. These findings suggest that the soil conditions play a major role in shaping the functional potential of the microbial communities, indicating that tailored field site management is a prerequisite for optimizing the benefits of microbial inoculants in improving microbial community functions and dryland soil health.

This study focused on the influence of microbial inoculants on the pea rhizosphere soil microbial communities using high-throughput sequencing. Field assessment of mycorrhizal colonization was not conducted due to challenges associated with precise root sample collection and potential inaccuracies stemming from confounding variables inherent in the field conditions. Despite this limitation, our focus on the broader rhizosphere microbiome provides a robust alternative in elucidating both bacterial and fungal community composition and diversity with higher resolution from phylum to genus taxonomic levels, which enables us to characterize the microbial communities of the untreated control (native microbiome) vs. the microbial treated (AMF, Rhizobium, AMF+Rhizobium). The microbial inoculants are beneficial root symbionts and widespread soil microorganisms [73,82]. These inoculants, applied as seed treatment, have a niche advantage that could influence the microbial community directly or indirectly [27,28]. However, in optimal environments, established plant-microbiome equilibrium may likely resist the influence of external inoculants due to low competitive pressure [83]. Noteworthy, we found a site-specific microbiome variation in response to microbial inoculation, which resulted in either a shift in microbial community composition or enrichment of beneficial soil microbes in sub-optimal dryland conditions. The extent of the microbial inoculants’ influence to cause a significant impact on microbial community functions relative to various agricultural management practices and other abiotic and biotic stresses in the drylands necessitates further studies. Our in situ field study provides insights into the overall microbial diversity and potential microbial functions crucial for understanding soil health and productivity in dryland agroecosystems.

## 5. Conclusions

Our study provides valuable insights into the influence of AMF and Rhizobium inoculants on plant yield, C sequestration, nutrient dynamics, microbial diversity, and the potential microbial community functions between contrasting dryland field sites using a high throughput sequencing approach. Our findings revealed that the efficacy of microbial inoculants is highly site-dependent. Across dryland sites, plant microbial dependency was higher in AMF+Rhizobium inoculation than in single inoculations. While microbial inoculants offer benefits in dryland conditions, their effectiveness is limited in high soil fertility, low organic matter, and acidic conditions. Microbial community dynamics exhibited site-specific variation, which showed a distinct microbial community pattern and a shift in bacterial and fungal species abundance between sites. The soil conditions showed a profound impact on potential microbial functions relative to nutrient cycling and ecophysiological functions. Additional follow-up studies will need to examine the influence of microbial inoculants on low-input field management, qPCR quantification of microbial inoculants, and validation of microbial community functions. Future efforts should focus on developing innovative crop and soil management strategies tailored to specific drylands’ unique characteristics and challenges in order to improve the overall dryland soil health and productivity.

## Figures and Tables

**Figure 1 microorganisms-12-00667-f001:**
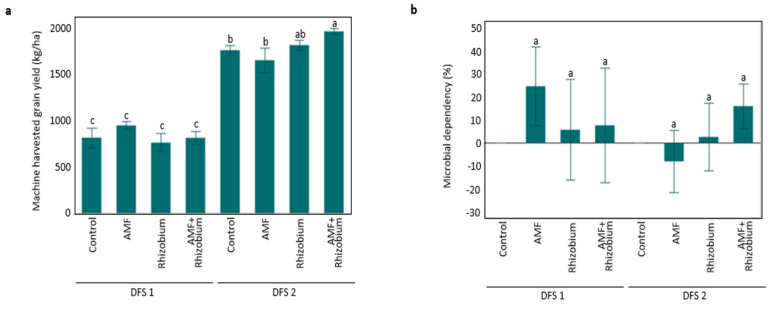
The influence of microbial inoculants on (**a**) plant grain yield and (**b**) microbial dependency (%) based on plant biomass and grain yield at contrasting dryland site conditions. Error bars indicate standard error of the mean from five replications. Bars with common letters are not significantly different based on Wilcoxon tests at 0.05% probability level.

**Figure 2 microorganisms-12-00667-f002:**
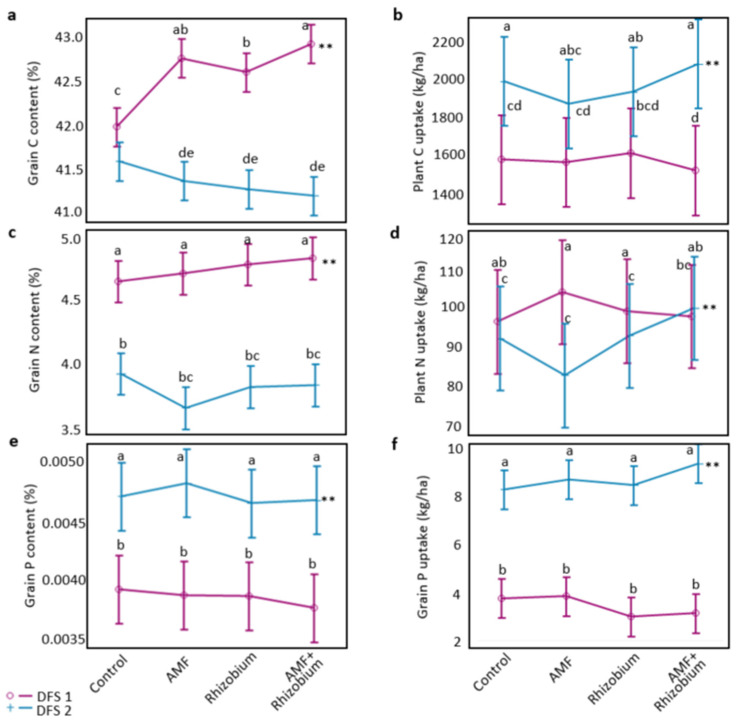
The effect of microbial inoculants on grain nutrient content (%) and plant nutrient uptake (kg/ha): carbon (**a**,**b**), nitrogen (**c**,**d**), and phosphorus (**e**,**f**) at two dryland field sites. The vertical bars in the least square means denote confidence intervals. Lines with common letters are not significantly different based on LSD tests at 0.05% probability level. Asterisk indicates dryland field site with significantly higher grain nutrient content and plant nutrient uptake, ** denotes significance level at *p* ≤ 0.001.

**Figure 3 microorganisms-12-00667-f003:**
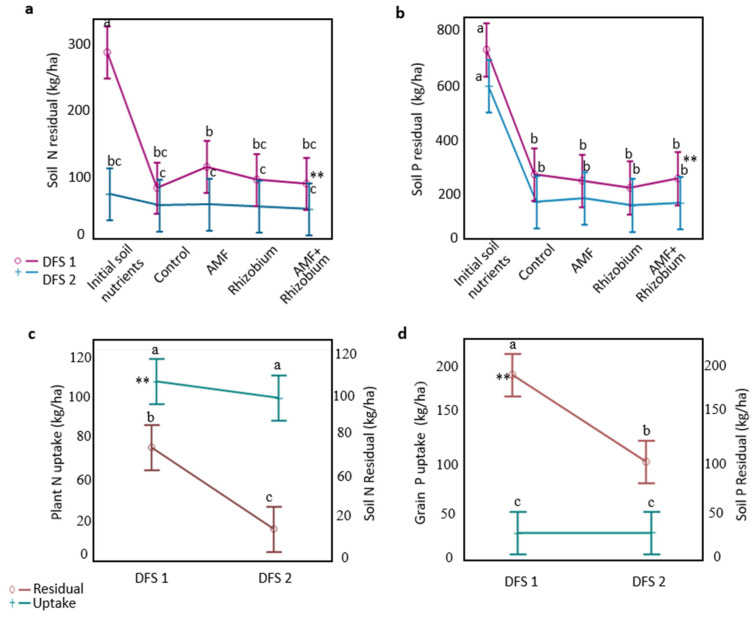
Effect of microbial inoculants on nutrient dynamics at contrasting dryland sites. Comparison of the initial soil nutrients vs. the effect of microbial inoculants on soil nutrient residuals after pea cropping between sites: soil N (**a**) and soil P residuals (**b**). Comparison of the plant nutrient uptake vs. the soil nutrient residuals across sites: plant N uptake vs. soil N residual (**c**) and grain P uptake vs. soil P residual (**d**). The vertical bars in the least square means denote confidence intervals. Lines with common letters are not significantly different based on LSD tests at 0.05% probability level. Asterisk indicates dryland field site or comparison between plant nutrient uptake and soil nutrient residual with significantly high nutrient levels, ** denotes significance level at *p* ≤ 0.001.

**Figure 4 microorganisms-12-00667-f004:**
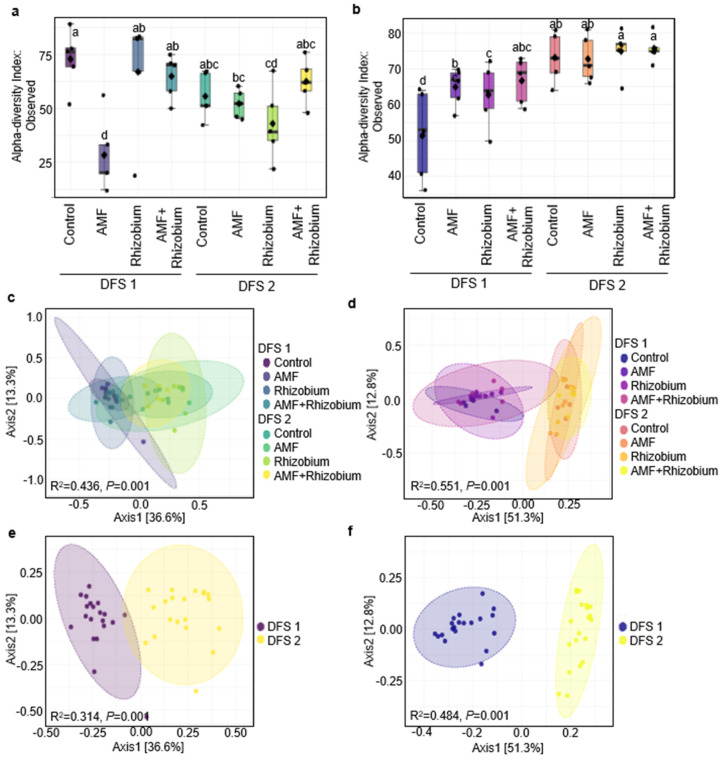
Site-specific effect response to microbial inoculation on microbial diversity. Alpha-diversity of bacterial (**a**) and fungal communities (**b**) was calculated as observed number of species per sample and visualized using box-plots. Beta-diversity of microbial communities among treatments at the two sites for the bacterial (**c**) and fungal communities (**d**), and between site comparisons for bacterial (**e**) and fungal communities (**f**). Beta diversity was calculated using the Bray–Curtis index and visualized using principal coordinate analysis (PCoA) ordination plots. The different groups are highlighted by ellipses showing a 95% confidence range and colored areas correspond to the bacterial and fungal community structure of the different treatments and sites.

**Figure 5 microorganisms-12-00667-f005:**
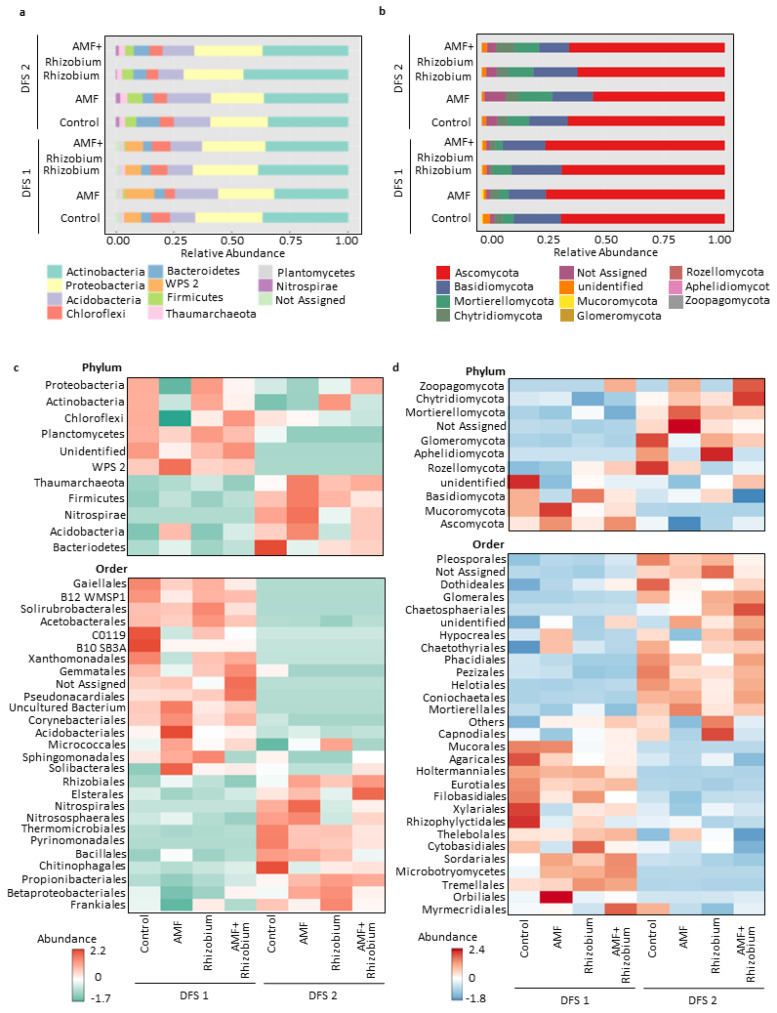
Taxabar plots showing the microbial community profiles of bacterial (**a**) and fungal communities (**b**) at the phylum level of the different treatments. Heatmap showing the microbial community pattern at the phylum and order taxonomic level composition of the bacterial (**c**) and fungal communities (**d**) at two dryland field conditions.

**Figure 6 microorganisms-12-00667-f006:**
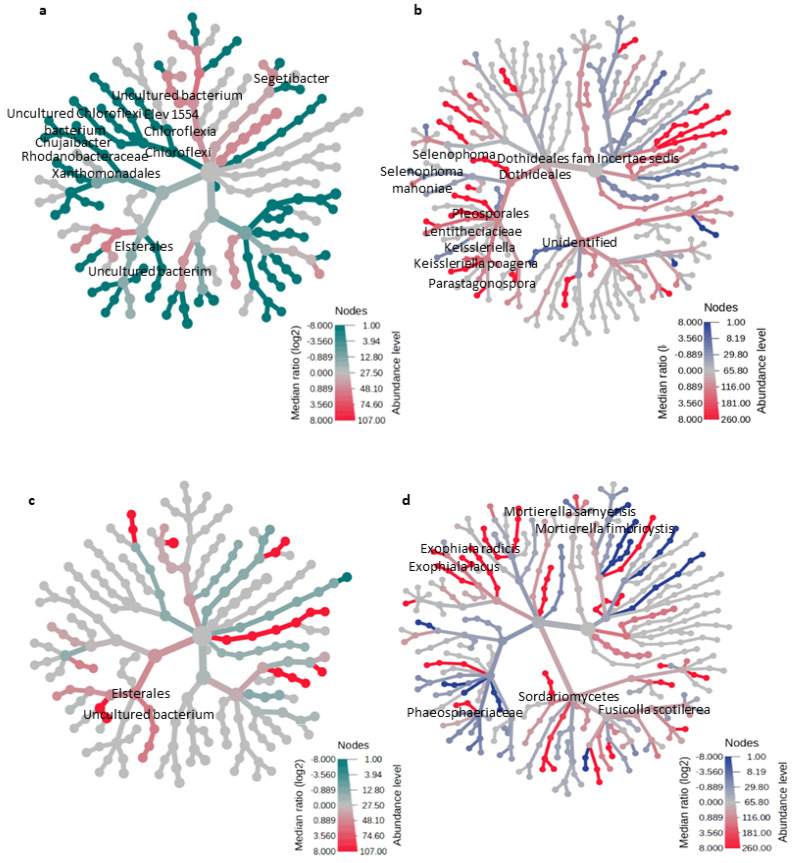
The heat tree showing the microbial communities of AMF and AMF+Rhizobium associated with increased crop performance in DFS 1 and DFS 2, respectively. The taxonomic differences between AMF-treated bacterial (**a**) and fungal communities vs. the control (**b**) in DFS 1; and AMF+Rhizobium-treated bacterial (**c**) and fungal communities vs. the control (**d**) in DFS 2. The heat tree analysis leverages the hierarchical structure of taxonomic classifications quantitatively using the median abundance and statistically using the non-parametric Wilcoxon Rank Sum test [56]. The indicated taxa with red nodes were significantly abundant in the microbial-treated plants, while green and blue nodes were significantly sparse in the bacterial and fungal communities of the microbial-treated plants compared to the untreated control.

**Figure 7 microorganisms-12-00667-f007:**
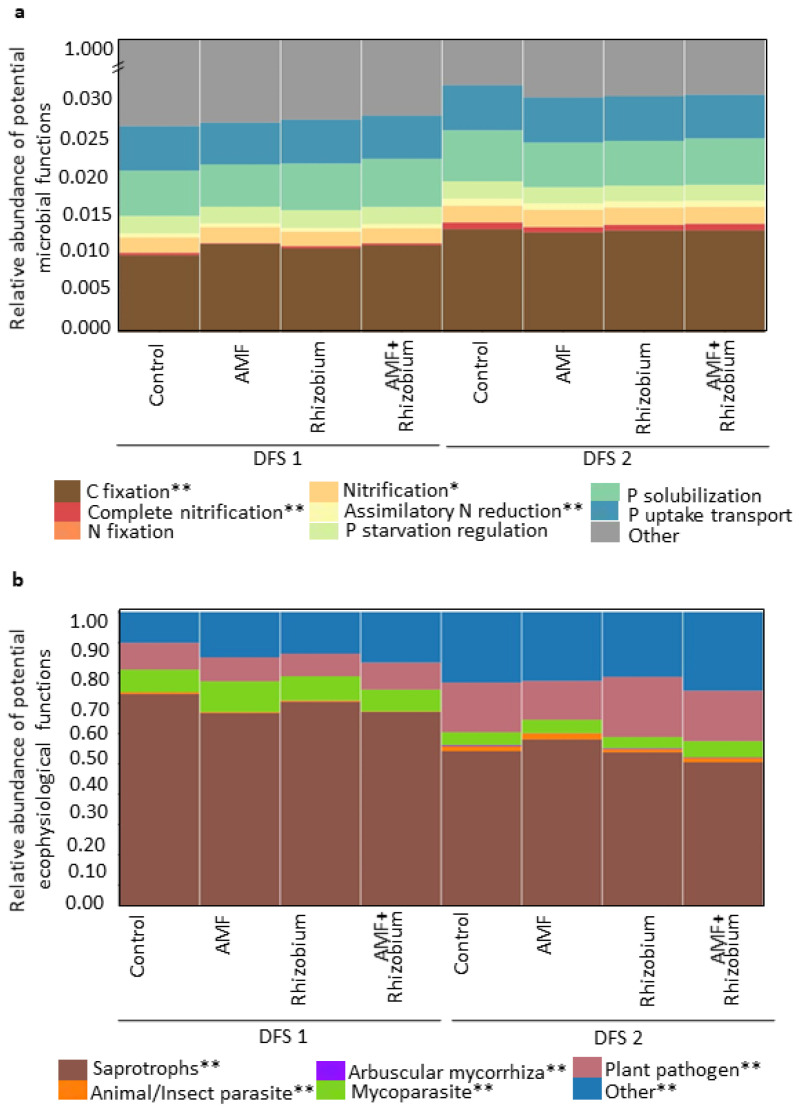
Impact of microbial inoculants on the relative abundance of potential microbial functions: functional profile of bacterial communities relative to C, N, and P nutrient cycling genes (**a**) predicted using Tax4Fun2 based on the 16S rRNA genes according to the KEGG Ortholog groups (KOs). Ecophysiological functions of fungal communities (**b**) relative to nutrient cycling, plant-microbe interaction, and soil health based on the FungalTraits database. Asterisk indicates microbial function with signficant difference between sites. * and ** denote significance levels at *p* ≤ 0.05 and *p* ≤ 0.001, respectively.

## Data Availability

The sequence data analyzed in this study were deposited in the NCBI database, BioProject PRJNA1020395 (https://www.ncbi.nlm.nih.gov).

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
