# Peer review of "Arbuscular Mycorrhizal Fungi and Rhizobium Improve Nutrient Uptake and Microbial Diversity Relative to Dryland Site-Specific Soil Conditions"

_microorganisms, 2024, doi:10.3390/microorganisms12040667_

Round 1
Reviewer 1 Report (Previous Reviewer 3)
Comments and Suggestions for Authors
The authors present work on improving the condition of plants growing in unfavorable conditions of limited availability of nutrients and water.
Research shows that the environment significantly modifies the impact of microorganisms on plants. The results also show that the combined administration of AMF and Rhizobium produces a more pronounced effect than the administration of AMF or Rhizobium alone.
It is clear that if we want to improve the uptake of nutrients, we need to look at the environmental conditions to optimize the effect of inoculation with microorganisms.
The authors responded to all my comments. The inscriptions on the Figures (Fig 5) are slightly clearer and, although still not perfect, can be read.
The authors present a wide range of results and draw clear conclusions. Although the combined use of AMF and Rhizobium is not very innovative, in my opinion the work brings new elements and is suitable for publication.
Author Response
Dear Reviewer,
Thank you for your insightful comments and thoughtful inputs on our manuscript.
Reviewer 2 Report (Previous Reviewer 2)
Comments and Suggestions for Authors
The work still requires editing - please use a service to do this
The work points to the fact that seed treatments with microbes generally accepted as being beneficial for the plant also impact the other microbes associated with the inoculated plants. The impact of soil traits is also highlighted

the revised version with all the cross outs and changes on changes is difficult to read several sentences are not configured optimally
suggest they use a professional editor
I will not review again
Author Response
Dear Reviewer,
We appreciate your insightful comments and thoughtful inputs. Attached is the pdf file with few clarifications.
Thank you very much.

This manuscript is a resubmission of an earlier submission. The following is a list of the peer review reports and author responses from that submission.
Round 1
Reviewer 1 Report
Comments and Suggestions for Authors
The Manuscript ID microorganisms-2887713 entitled "
Arbuscular mycorrhizal fungi and rhizobium improve nutrient uptake and microbial diversity relative to dryland site-specific soil conditions" The study is interesting. However, I suggest it needs a minor revision to be published in this journal. I do have some comments about the manuscript and data interpretation/discussion that could improve the overall quality of the manuscript.
Abstract : The abstract is well-written; however, it lacks quantitative information regarding the results achieved by the researchers.
Introduction : - The introduction is excellent; however, it requires a graph detailing the pea plant used in the study, its economic importance, and the risks it faces.
- You need a paragraph that discusses the use of fertilizers and biocides as therapeutic nutrition, their role in enriching the soil with nutrients and biodiversity, and their environmental impact. You may use these relevant references.(https://doi.org/10.3390/life13010012, https://doi.org/10.1007/s13399-023-05103-x, https://doi.org/10.1186/s40529-022-00364-7, https://doi.org/10.3390/jof8080775 ).
Materials and methods: there is some minor suggestions:
(line 107) : Please write the method in detail
Lines: 107:112: I suggest moving this section to the results section:
"The available soil nutrients were compared to the pea nutrient requirements (Table S3). DFS 1 had a strongly acidic soil (pH = 4.7), low organic 108
matter content (1.5% LOI), medium nitrogen (28.33 ppm N), very high phosphorus (73.00 ppm P), and potassium (214.5 ppm). Meanwhile, DFS 2 had a slightly neutral pH (pH= 6.33), moderate organic matter content (2.5% LOI), low nitrogen (7.60 ppm N), high phosphorus (44.75 ppm P), and very high potassium (277.87 ppm K)."
Results :
Figure 4: needs to more clarity.
(lines: 355:357) : I suggest moving this section to the material and methods section.
Figure 6: needs to more clarity.
Discussion : Wonderfully written.
Conclusion: It comprehensively explains the study's achievements but needs to be shortened.
References :
- No self-citation except 2 only .
- The references are recent and consistent with the study.
Author Response
Reviewer 1
- Abstract: The abstract is well-written; however, it lacks quantitative information regarding the results achieved by the researchers. - The contribution of the different treatments to the overall pant growth and yield based on microbial dependency (%) was indicated in the abstract.
- Introduction: - The introduction is excellent; however, it requires a graph detailing the pea plant used in the study, its economic importance, and the risks it faces.
- You need a paragraph that discusses the use of fertilizers and biocides as therapeutic nutrition, their role in enriching the soil with nutrients and biodiversity, and their environmental impact. You may use these relevant references.(https://doi.org/10.3390/life13010012, https://doi.org/10.1007/s13399-023-05103-x, https://doi.org/10.1186/s40529-022-00364-7, https://doi.org/10.3390/jof8080775 ).
-Done
- Materials and methods: there is some minor suggestions: (line 107) : Please write the method in detail
- Done
- Lines: 107:112: I suggest moving this section to the results section:
"The available soil nutrients were compared to the pea nutrient requirements (Table S3). DFS 1 had a strongly acidic soil (pH = 4.7), low organic 108 matter content (1.5% LOI), medium nitrogen (28.33 ppm N), very high phosphorus (73.00 ppm P), and potassium (214.5 ppm). Meanwhile, DFS 2 had a slightly neutral pH (pH= 6.33), moderate organic matter content (2.5% LOI), low nitrogen (7.60 ppm N), high phosphorus (44.75 ppm P), and very high potassium (277.87 ppm K)."
-Done
- Results: Figure 4: needs to more clarity. -Done
- (lines: 355:357) : I suggest moving this section to the material and methods section. - Done. Lines 214-217
- Figure 6: needs to more clarity. - Done
- Discussion : Wonderfully written. -Thank you
- Conclusion: It comprehensively explains the study's achievements but needs to be shortened. -Done
Reviewer 2 Report
Comments and Suggestions for Authors
The work involves real field studies which is essential for understanding how most of the crops are grown currently.
The use of AMF and Rhiz is not novel nor is it a surprize that these were not shown to have clear cut results or that different soils gave different data sets-
For me there is an absolute need to have information on AM colonization and nodulation of the peas - this is the work site for the pea yeild and so the soil data which make up the bulk of the results is on the periphery
the methods lack details
A major problem is the speculation that it is the live microbes added as inoculant that are changing the soil microbe composition - no evidence provided since those inoculants are not traced at all
It looks as if at least there are abundant AMF in the two soils without inoculation
what else is in the inoculants?

There are many strange sentence structures
Notes have been made where the constructions could be improved
So yes editing is needed
Author Response
Reviewer 2
The work involves real field studies which is essential for understanding how most of the crops are grown currently.
- The use of AMF and Rhiz is not novel nor is it a surprise that these were not shown to have clear cut results or that different soils gave different data sets - While it is true that the use of AMF and Rhizobium is not novel concept in agriculture, our study focuses specifically on their application in dryland conditions, where their efficacy is not well-defined. Our aim is to contribute to the understanding of the applicability of AMF and Rhizobium in dryland agroecosystems, where their potential benefits have not been extensively studied. The variability observed between dryland sites indicates the importance of site-specific conditions and thus the varying efficacy of the microbial inoculants in shaping rhizosphere microbial communities that impacts crop production. This variability is a key aspect of our findings and highlights the need for tailored approaches in promoting sustainable agriculture in diverse dryland environments.
- For me there is an absolute need to have information on AM colonization and nodulation of the peas - this is the work site for the pea yield and so the soil data which make up the bulk of the results is on the periphery -We affirm the importance of mycorrhizal colonization and nodulation. We attempted to collect uniform root samples from the field, however, we observed confounding variations posed by field conditions. To reduce these variations, we used a rating scale relative to the entire plant root collected rather than counting the root nodules per plant, nonetheless, nodulation failure was observed in DFS 1, while in DFS 2 an increasing trend was observed but there was no significant differences in the nodulation. In the mycorrhizal colonization, we plan to conduct a different experiment setup under controlled condition to avoid potential inaccuracies stemming from the challenges associated with field-based AMF colonization assessments.
- the methods lack details. -More details included in the methodology.
- A major problem is the speculation that it is the live microbes added as inoculant that are changing the soil microbe composition - no evidence provided since those inoculants are not traced at all. -Our study utilized high-throughput sequencing of the 16S (bacteria) and ITS rRNA (fungi) marker genes, which provides a high resolution identification of the microbial community compositions from different taxonomical levels (phylum to genus). While we acknowledge that tracing live inoculants directly was not part of our methodology, the choice of bacterial and fungal marker genes allows us to distinguish closely related taxa to the microbial inoculants used in the study. The rhizosphere microbiome analysis enable us to characterize the microbial communities of the untreated control (native microbiome) vs. the microbial treated (AMF, Rhizobium, AMF+Rhizobium), which provides insights into the overall microbial diversity and potential microbial functions crucial for understanding the soil health and productivity of dryland agroecosystems. We acknowledge the importance of incorporating additional tracing techniques in future studies to directly follow the fate of live inoculants in the soil.
- It looks as if at least there are abundant AMF in the two soils without inoculation -Based on the ITS rRNA sequence data, Glomus/Rhizophagus genera were abundant in AMF and AMF+Rhizobium treatments in DFS 2. Although, the Glomeromycota phylum (with two classes: Glomeromycetes and Paraglomeromycetes) was significantly higher in the untreated control in DFS 2, indicating the abundance of other native arbuscular mycorrhizal fungi. Within DFS2, the Glomerales order (which includes Glomus/Rhizophagus and other AMF genera: Claroideoglomus, Dominikia, Funneliformis, and Septoglomus) had high relative abundance in the AMF+Rhizobium-treated.
- What else is in the inoculants? - The commercial microbial inoculants are formulated in peat-based powder, which are the common carrier known to support the growth of various microorganisms. While, this carrier could introduce additional organic material, it may not be substantial enough to cause soilphysicochemical changes, and its potential impact likely affect all soil microbes rather than selectively.
- peer-review-35189172.v1.pdf. Comments on the Quality of English Language. There are many strange sentence structures. Notes have been made where the constructions could be improved. So yes editing is needed. -The comments in the pdf file were incorporated and sentence structures were corrected.

Reviewer 3 Report
Comments and Suggestions for Authors
The authors analyzed the effect of adding AMF or AMF+Rhizobium to the soil at two different field sites in dry areas. The soils from these two sites differ in physicochemical terms.
The authors presented an analysis of the impact of microorganisms on the yield of peas and the uptake of nutrients from the soil.
Further analysis show changes in the microbiome after inoculation with AMF or AMF+Rhizobium. The authors also analyze in situ the potential activity profile of bacteria in the soil and the potential functions of fungi.
Authors noticed higher yield of plants inoculated with AMF+Rhizobium and lower to these inoculated only by AMF in the DFS2 and very high plant growth and yield (25%) in the DFS1 (low soil organic matter and low pH) treated by AMF. Plant growth was also increased by AMF+ Rhizobium and AMF or Rhizobium in DFS1 location.
Next the authors measured nutrients uptake after treatment with microorganisms in both locations. The conclusions are that microbial inoculation has less influence on nutrient uptake compared to the initial concentration of them in the soil.
Alpha diversity showed that DFS2 richer in soil organic matter displayed higher fungal and bacterial diversity and inoculation by AMF and AMF+Rhizobium elevated the fungal diversity while bacterial alpha diversity decreased in the DFS1. Authors concluded that the site specific conditions strongly influenced the microbial community structure.
I think the work is valuable, but the quality of some figures needs to be improved.
My comments:
Line 315 “AMF+Rhizobium had significant enrichment of Elsterales.” Rather soil treated by AMF+Rhizobium had significant enrichment …..
Line 318 – “were abundant” non italic
Please correct the quality of Fig 3, 5- the inscriptions on the figures are illegible
Fig 4 Microbes phyla and orders
In both sites after microbial treatment by Rhizobium which belongs to the alpha proteobacteria on Fig 4a the relative abundance for Proteobacteria are the same as in the control. What does it means?
The same for fungi. The locations are treated by AMF Glomus intraradices. Have you seen the increase of the Glomeromycota in both locations? (Fig 4 b and d)
Fig. 4 c and d
For better understanding of the data and connection with Fig. 4 a and b, information about the phylum should be added to graphs c and d. This information will allow the reader to coordinate information from parts a and b with c and d and assign orders to the phylum.
Fig.5 the description in the drawings is very poorly visible and must be corrected
Author Response
Reviewer 3
I think the work is valuable, but the quality of some figures needs to be improved.
- The figures were improved.
My comments:
- Line 315 “AMF+Rhizobium had significant enrichment of Elsterales.” Rather soil treated by AMF+Rhizobium had significant enrichment ….. -Done
- Line 318 – “were abundant” non italic -Done
- Please correct the quality of Fig 3, 5- the inscriptions on the figures are illegible -Done
- Fig 4 Microbes phyla and orders. In both sites after microbial treatment by Rhizobium which belongs to the alpha proteobacteria on Fig 4a the relative abundance for Proteobacteria are the same as in the control. What does it means? -The Proteobacteria is the largest bacterial phylum, which includes a wide range of bacteria from beneficial to harmful groups. The beneficial class of Alphaproteobacteria were abundant in the microbial inoculated treatments with AMF+Rhizobium in DFS2.
The same for fungi. The locations are treated by AMF Glomus intraradices. Have you seen the increase of the Glomeromycota in both locations? (Fig 4 b and d) -The Glomeromycota phylum was significantly higher in DFS2 based on relative abundance. Within DFS2, the Glomerales was slightly higher in the AMF+Rhizobium-treated. The relative abundance of Glomus was slightly higher in the untreated control attributed to its low fungal diverstiy. While, AMF and AMF+Rhizobium treatments had higher diversity, thus, a lower relative abundance of Glomus, eventhough there was a higher number of observed sequences of Glomus in the AMF and AMF+Rhizobium treated plant rhizosphere microbiome.
- F 4 c and d
For better understanding of the data and connection with Fig. 4 a and b, information about the phylum should be added to graphs c and d. This information will allow the reader to coordinate information from parts a and b with c and d and assign orders to the phylum.
- Done
- 5 the description in the drawings is very poorly visible and must be corrected -Done